# Encoding of cerebellar dentate neuron activity during visual attention in rhesus macaques

**Nico A Flierman[1,2†], Sue Ann Koay[3†], Willem S van Hoogstraten[2], Tom JH Ruigrok[2], Pieter Roelfsema[1,4,5], Aleksandra Badura[2\*], Chris I De Zeeuw[1,2\*]**

[1]Netherlands Institute for Neuroscience, Amsterdam, Netherlands; [2]Department of Neuroscience, Erasmus MC, Rotterdam, Netherlands; [3]Janelia Research Campus, Howard Hughes Medical Institute, Ashburn, United States; [4]Department of Integrative Neurophysiology, VU University, Amsterdam, Netherlands; [5]Department of Psychiatry, Academic Medical Centre, Amsterdam, Netherlands

## eLife Assessment

This **important** study examined neuronal activity in the dentate nucleus of the cerebellum when monkeys performed a difficult perceptual decision-making task. The authors provide **convincing** evidence that the cerebellum represents sensory, motor, and behavioral outcome signals that are sent to the attentional system. This paper is of great general interest in that it shows the involvement of the cerebellum in cognitive processes at the neuronal level.

**\*For correspondence:**
a.badura@erasmusmc.nl (AB);
c.dezeeuw@erasmusmc.nl (CIDZ)

[†]These authors contributed equally to this work

**Competing interest:** The authors declare that no competing interests exist.

**Abstract** The role of cerebellum in controlling eye movements is well established, but its contribution to more complex forms of visual behavior has remained elusive. To study cerebellar activity during visual attention we recorded extracellular activity of dentate nucleus (DN) neurons in two non-human primates (NHPs). NHPs were trained to read the direction indicated by a peripheral visual stimulus while maintaining fixation at the center, and report the direction of the cue by performing a saccadic eye movement into the same direction following a delay. We found that single-unit DN neurons modulated spiking activity over the entire time course of the task, and that their activity often bridged temporally separated intra-trial events, yet in a heterogeneous manner. To better understand the heterogeneous relationship between task structure, behavioral performance, and neural dynamics, we constructed a behavioral, an encoding, and a decoding model. Both NHPs showed different behavioral strategies, which influenced the performance. Activity of the DN neurons reflected the unique strategies, with the direction of the visual stimulus frequently being encoded long before an upcoming saccade. Moreover, the latency of the ramping activity of DN neurons following presentation of the visual stimulus was shorter in the better performing NHP. Labeling with the retrograde tracer Cholera Toxin B in the recording location in the DN indicated that these neurons predominantly receive inputs from Purkinje cells in the D1 and D2 zones of the lateral cerebellum as well as neurons of the principal olive and medial pons, all regions known to connect with neurons in the prefrontal cortex contributing to planning of saccades. Together, our results highlight that DN neurons can dynamically modulate their activity during a visual attention task, comprising not only sensorimotor but also cognitive attentional components.

## Introduction

The crucial role of the cerebellum in sensorimotor processing, such as learning and execution of eye movements, is well established (*Inoshita and Hirano, 2018*; *Ivry and Diener, 1991*; *Kunimatsu et al., 2016*). Yet, over the last few decades evidence is emerging that the cerebellum also participates in more complex cognitive visual functions (*Baier et al., 2010*; *Brissenden et al., 2016*; *Courchesne et al., 1994*; *Nicolson et al., 2001*; *Voogd et al., 2012*). For example, while interacting with areas such as the frontal eye fields (FEFs), lateral intraparietal area, superior colliculus, and basal ganglia, the cerebellum uses visual information to guide the planning of saccades (*Gao et al., 2018*; *Kunimatsu et al., 2018*; *Scerra et al., 2019*; *Tanaka, 2006*; *Wardak et al., 2011*). An open question is to what extent the cerebellum also has access to information in the peripheral visual field to adjust the planning of saccades. If the cerebellum is prominently involved in such processing, one expects to find encodings of both visual cues and the related eye movements at its output stage in the cerebellar nuclei.

In daily life, visual scenes often contain considerably more information than the visual system can process in short periods of time. Therefore, saccades are made to align the fovea, the region of the retina having the highest visual acuity and largest representation in the visual cortex, with the target of interest. Yet, it remains to be elucidated how saccades are planned. Neuroimaging studies suggest that network mechanisms in the parietal, frontal, and temporal lobes that control the programming of saccades overlap with those underlying covert attention (*Corbetta et al., 1998*; *Nobre et al., 2000*). Along the same avenue, the 'premotor theory of attention' by *Rizzolatti et al., 1987* proposes that the circuit used for generating movements is also active during attention shifts. This theory is also supported by the observation that the working peripheral location of the covert cue stimulus is restricted by the ultimate potential outer position of the eye in that it is not possible to attend to a covert cue when the cue is outside of the range of the oculomotor system (*Craighero et al., 2004*; *Hanning and Deubel, 2020*). At the same time, it should be noted that when it comes to neuronal responses at the single-cell level, the cortical networks engaged in covert attention do not always perfectly overlap with those that control saccades. For example, findings in the FEF show that at least part of the visual cells that respond to covert attention do not modulate in relation to the saccades, implying that there may be cell type – specificity for responses to covert attention in this region (*Gregoriou et al., 2012*).

In this work, we aimed to examine if visual attention stimuli are processed in the dentate nucleus (DN) of the cerebellum, and to what extent the signaling during a visuomotor task controls the directional saccadic eye movements. We tested the overall hypothesis that DN neurons can dynamically modulate their activity during a covert visual attention task, comprising a mixture of sensorimotor and cognitive attentional, that is, multimodal, components. More specifically, given that earlier studies in primates and humans *Herzfeld et al., 2018*; *van Es et al., 2019* have indicated that direction selectivity of cerebellar activity may occur in various sensorimotor domains, we hypothesized that DN cells can be direction selective for not only the sensorimotor but also the cognitive components. Therefore, we studied the activity of single units in the DN during a complex, peripheral Landolt C saccade task in two non-human primates (NHPs). The task comprised the following consecutive components: (1) shifting attention toward the C-location in the peripheral visual field, while holding gaze fixation on the center (*Figure 1A*), (2) recognizing of the location of a 'gap' in a virtual full circle as eminent in the letter C, which provides the cue for the direction of the saccade to be made in the future (while still holding central gaze fixation), (3) planning of a saccade into the direction that depends on the peripheral visual information (i.e., position of the gap in the C), (4) executing the corresponding saccade following a delay, and (5) receiving a reward if the direction of the saccade is correct. We found that DN neurons could modulate their activity during multiple task epochs, often selectively for an individual C-gap direction or saccade direction. After analyzing DN activity and choice performance following task relearning, it became evident that each animal employed a different behavioral strategy during execution of the task. In addition, early onset of stimulus triggered activity was detected in the better performing animal. Finally, the location of the task-relevant neurons in the DN was in line with the known connections from and to the FEF, that is, one of the main classical attention and saccade planning center(s) in the cerebral cortex (*Gregoriou et al., 2012*).

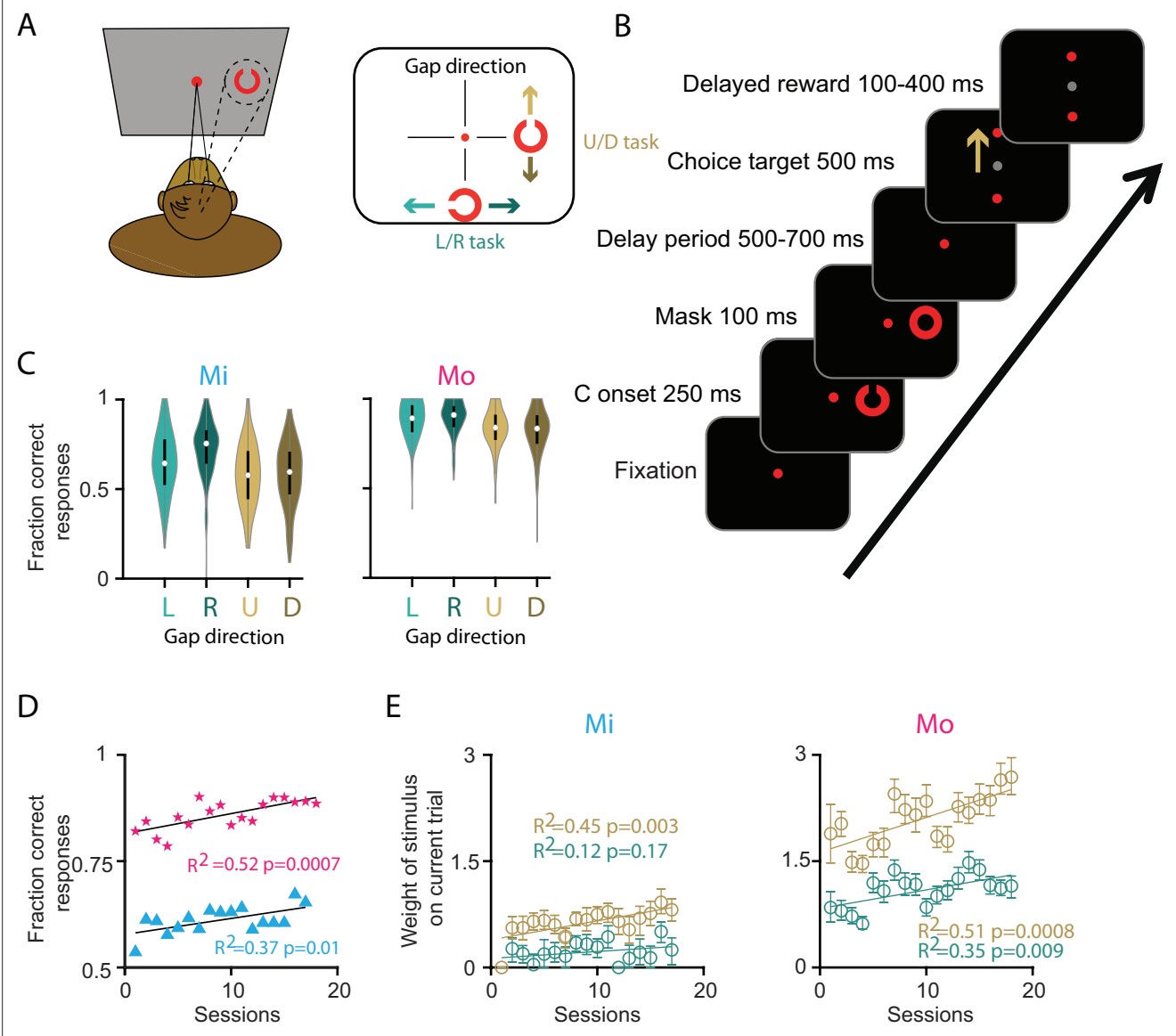

**Figure 1.** Task and behavioral performance. (**A**) Left, drawing of the animal performing the task. Right, schematic overview of directionality in the task: only one C-stimulus is presented per trial in one of four cardinal directions from the central fixation dot. This could be to the top/bottom of the central fixation, with possible gap directions to the left and right in the [L/R] task, or upward and downward in the [U/D] task. (**B**) Example trial with a C-stimulus presented to the right, with an 'Up' gap direction, where the animal has to make an upward saccade. Delay period between the mask and the onset of the target varied between 500 and 700 ms (randomly drawn from 500, 550, 600, 650, and 700 ms). The saccade had to be performed in the epoch between the appearance of the choice target and the reward. Beige arrow indicates the correct choice, not actually visible to the monkey. (**C**) Overview of performance per animal per gap direction. Black bar at the middle of the violin plot represents the 25th and 75th percentile, the white dot at the center is the median. (**D**) Average fraction of correct responses per retraining session, after the animal had had several weeks of 'holiday'. Each session represents the median performance of 1 day. (**E**) Weight, that is, relative impact, of the gap direction of the C-stimulus in predicting the correct response through logistic regression. Each point is the average of one session, error bars indicate 95% confidence interval from 100 bootstrap permutations. Beige data points and fit represent the up/down task, the green data the left/right task.

The online version of this article includes the following figure supplement(s) for figure 1:

**Figure supplement 1.** Weight, that is, relative impact, of the gap direction of the C-stimulus of the previous trial to predict the response on the next trial over time; the trials could be from either an L/R or a U/D task, hence the two conditions per panel.

# Results

Two male rhesus macaques (Mo and Mi) were trained to perform a peripheral Landolt C task (*Ignashchenkova et al., 2004*; *Figure 1A*), allowing us to assess how DN cells encode visual stimuli and related (preparatory) motor activity. The animals had to maintain central fixation throughout the trial as long as the central fixation dot remained red (see example trial in *Figure 1B*). After a delay of ~500 ms after fixation onset, the C-stimulus was randomly presented in one of four cardinal directions (Right = 0°, Left = 180°, Up = 90°, Down = 270°) for the duration of 250 ms. The animals had to perceive the direction of the gap of the C (gap direction), which was always positioned perpendicular to the stimulus location. Specifically, the gap direction is either Up or Down [U/D] when the C-stimulus was presented in locations L or R [L/R], or the gap direction is either Left or Right [L/R] when the C-stimulus was presented in locations U or D. After the presentation of the C-stimulus, the C was masked for 100 ms to prevent retinal afterimages, and subsequently there was a delay period of 500–700 ms in which Mi and Mo had to remember the gap direction. After the delay period, two saccade targets were shown at the same eccentricity; one target was placed in the same direction as the gap direction (i.e., the correct target), and the other one was presented as a distractor. If the animals made a saccade to the correct target within 500 ms after the target onset, they received a juice reward.

## Task and behavior

There were significant differences in task performance levels between Mi and Mo that did not depend on the gap direction (*Figure 1C*). Mo consistently outperformed Mi, and this trend was present across different experimental sessions (*Figure 1D*). It should be noted that this analysis used data from periods after the monkeys had been sufficiently trained to understand the task structure above-chance level (i.e., >60% correct responses), so the performance differences do not directly represent the structure of the initial learning process. Still, since for each animal the task sessions were separated by weeks-long breaks, during which the other animal was subjected to electrophysiological recordings, some level of retraining likely took place when sessions restarted.

To assess which task variables contributed to the behavioral performance in the individual animals we split the type of tasks in two sets: the upward and downward directions (U/D task) and the leftward and rightward directions (L/R task). This allowed us to apply a logistic regression model with the dependent variable being the fraction of correct responses, and the independent variables being those that specify the task structure. We first explored how the gap direction presented in the current trial was weighted to best predict the animals' performance over the sessions. The weight that Mi assigned to the U/D gap direction increased simultaneously with task performance. In contrast, the weight that was assigned to the L/R stimulus remained constant, suggesting that his overall performance increase was mostly mediated by his improvement in the subset of up-/downward trials in the task (*Figure 1E*, left panel). Mo, on the other hand, exhibited increased weighting of the stimulus in both U/D and L/R trials. Indeed, the overall higher performance of Mo relative to Mi was reflected in the model on Mo's data having higher weights for the gap direction throughout all measured sessions (*Figure 1E*, right panel). We next assessed if any past-trial conditions influenced the behavioral response in the current trial. Therefore, we determined the weight of events gap direction and saccade direction in the previous trial with the same logistic regression approach. According to this model, Mi assigned less weight to the previous saccade over time if the current trial was in the L/R direction (p = 0.016; $R^2$ = 0.32, *Figure 1—figure supplement 1*). In contrast, there was no change in event weight to predict the response on the current trial in the past-trial variables for Mo.

## DN activity

Once the animals reached an above-chance performance level (60%), we identified the site of the DN in both Mi and Mo with the use of structural MRI (*Figure 2A*). Single-unit activity was recorded for 305 DN neurons, while the animals performed the Landolt C task. The activity levels of DN neurons in both Mi and Mo changed during the instruction and movement stage, depending on the stimulus identity (*Figure 2B, D*; for separate datasets of Mi and Mo, see *Figure 2—figure supplement 1*). For example, during the presentation of the C-stimulus a DN cell could respond with an increase in activity, followed by suppression around the time of the correctly executed saccade (*Figure 2C*). Yet, the level of activity during the facilitation after the presentation of the C-stimulus could be selective for trials with different directions; for example, it could be higher for trials during which the gap direction was

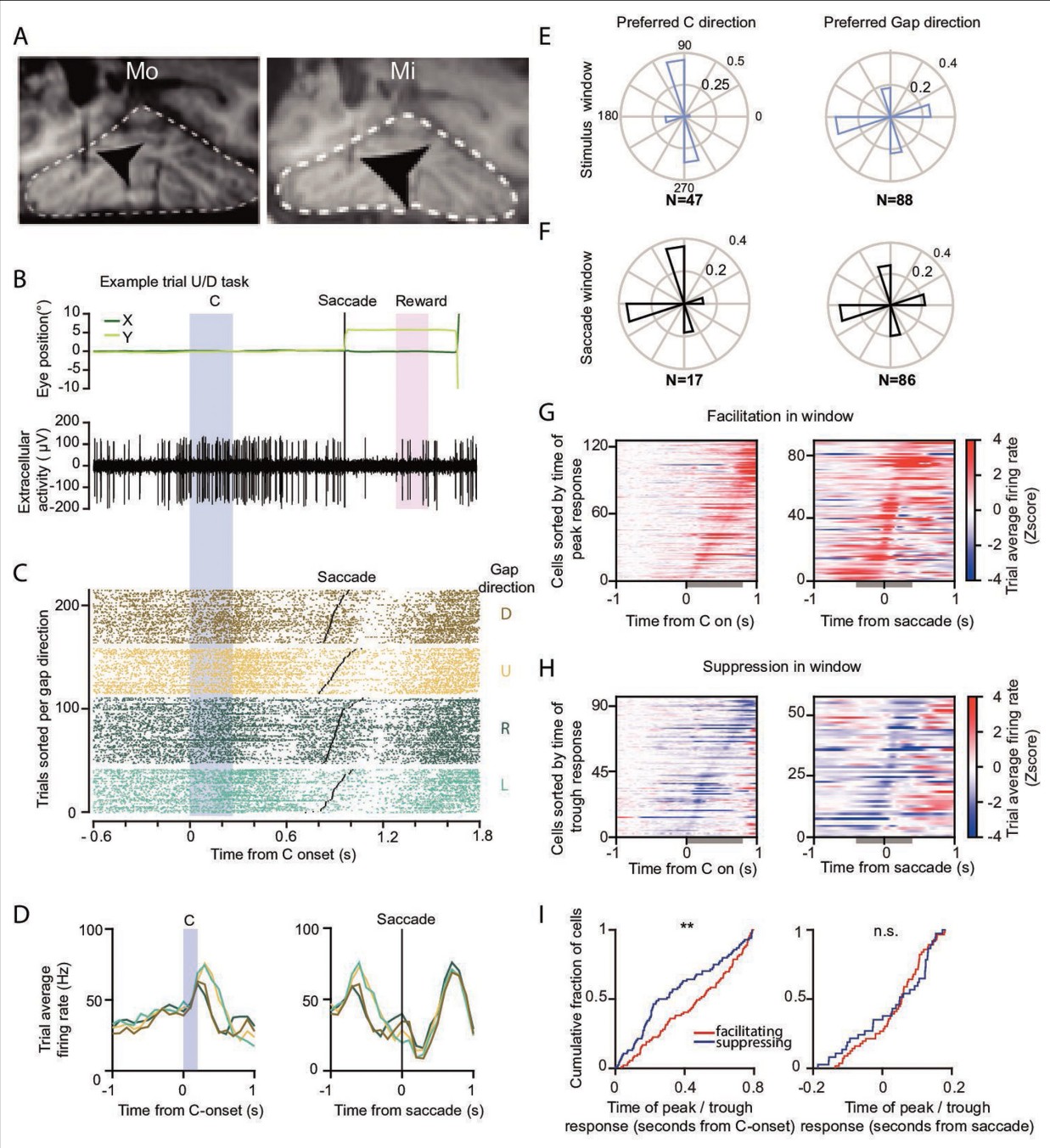

**Figure 2.** Neural activity in the dentate nucleus (DN) during the Landolt C task. (**A**) MRI images of Mi and Mo, showing the electrode position in the DN. (**B**) Example trial of extracellularly recorded DN cell in Mi aligned to C-stimulus onset (blue shaded area shows when C is visible), while the animal maintained fixation until the potential targets were presented (in this case an upward saccade was correct). Top panel depicts eye position, bottom panel shows the single-unit activity of the DN cell. Pink shaded area represents juice reward pulse duration. (**C**) Raster plot of correct trials sorted per gap direction, represented per color, of the cell presented in B. Within each direction, trials are sorted by the time of saccade onset (black dot in each trial). (**D**) Average activity per direction in 100 ms bins in spikes per second. Left panel displays activity aligned to C-stimulus onset, right panel displays activity aligned to saccade onset. (**E**) Preferred directions of the population of neurons. Normalized polar histograms show the fraction of cells that preferred the stimulus direction (left) or the action direction (right). (**F**) Same as E for direction preference during saccade epoch. (**G**) Normalized activity of cells displaying facilitating activity deviating from baseline in either the window after C-stimulus onset (gray bar, 0 to 800 ms, left panel) or around the saccade (gray bar, −200 to 200 ms, right panel). (**H**) Same as in G, but for suppressed cells. (**I**) Timing of peak/trough responses from G and H, stars ** indicate significance at p = 0.01 level, according to Kolmogorov–Smirnov test; ns = not significant. For separate datasets of Mi and Mo, see Figure S2.

The online version of this article includes the following figure supplement(s) for figure 2:

**Figure supplement 1.** Neural activity in the dentate nucleus (DN) during the Landolt C task separated for both animals.

either left or up, compared to those with right or down directions (*Figure 2D*). Differences in spiking for different directions were determined by comparing the number of spikes in a bin at the moment of largest divergence between directions (see Methods). Changes of neuronal activity in relation to the presentation of the C-stimulus and/or the saccade were common (*Figure 2E, F*). For example, cells could show significant facilitation of their activity in a window associated with the C-stimulus (200 ms moving window, 100 ms steps from 0 to 800 ms after C-stimulus onset) or the saccade (200 ms moving window, 100 ms steps from −200 ms before to 200 ms after saccade onset) relative to baseline activity from before trial onset (see Material and methods for details) (*Figure 2E*). Likewise, cells could also show suppression of their firing following the C-stimulus and the saccade (*Figure 2F*). When we aligned the neurons to the time of their maximum change in activity after C-stimulus onset, the population of suppressing neurons ($n = 85$) exhibited the modulation significantly earlier ($p < 0.0041$; Kolmogorov–Smirnov test) than did the facilitating units ($n = 90$), (*Figure 2G–I*). This was not the case ($p = 0.23$) for facilitating ($n = 57$) and suppressing ($n = 38$) cells during the saccade window (*Figure 2I*).

## Encoding of task events

Many DN cells modulated their activity in relation to more than one task event. To fully characterize the neural responses, we fitted a generalized linear model (GLM) and analyzed which of the task events were encoded in the spike activity (*Figure 3*). This model predicts the spiking activity time course of individual neurons as a weighted sum of time-dependent basis functions in response to every task event (*Park et al., 2014*). LASSO regularization was used to select a parsimonious set of task events that should be included in the model (*Figure 3A–C*; for details see Materials and methods). We found that many cells encoded the gap direction, the saccade or both, while a smaller fraction was sensitive to the events in the previous trial (*Figure 3B, C*). Many units showed firing rate modulation in relation to both particular gap directions and saccades (*Figure 3D, E*). Likewise, a neural response after the saccade in trials where the animal made a mistake was a common feature (*Figure 3—figure supplement 1*). Since retinal errors were not expected to be a relevant feature of this task, at least not in the sense of those that may occur in classic saccadic adaptation paradigms (*Herzfeld et al., 2018*), it is possible that these responses are related to the absence of reward (*Figure 3D, E*; see also *Heffley and Hull, 2019*; *Kostadinov et al., 2019*). Interestingly, these correlations could come with both facilitations and suppressions (*Figure 3D, E*). Despite the level of heterogeneity among the two NHPs in terms of performance level and strategy, the fractions of recorded neurons that were sensitive to particular task parameters, including gap direction, saccade direction, or miscellaneous were similar in both animals, indicating that we sampled from comparable populations of DN neurons in the two NHPs (*Figure 3C*).

Principal component analysis (PCA) of activity during correct and incorrect trials shows differences in latencies of the ramping activity following presentation of the visual stimulus and may contribute to the differences in performance between Mi and Mo (*Figure 3—figure supplement 1*). We analyzed the ramping of firing rates in relation to the moment of the C-stimulus presentation (*Satopaa et al., 2011*), and found that the latency from the onset of the C-stimulus to the moment of ramping in Mo was on average 115 ms shorter ($F = 10.9$, $p = 0.001$) than that of Mi (*Figure 3—figure supplement 2*).

## Decoding trial variables

Considering the heterogeneity of the neural activity in relation to the task, we built a decoding model to reveal which parameters of the task could reliably be predicted from the spiking activity of individual cells. For each 200-ms time window in the trial, we fit a logistic regression model to determine at which time points a given trial condition could be decoded from the cell's activity. The condition that was most reliably represented was the trial outcome, that is, whether the monkey made a correct or incorrect choice (*Figure 4A*). The proportion of cells from which the trial outcome could be decoded increased at about 1 s following the presentation of the C-stimulus to the monkey, and this persisted throughout the delay period and remained present for more than a second after the monkey had made his choice (*Figure 4A*, see different columns). Compared to Mo, Mi had more cells from which the trial outcome could be decoded around the time of the reward. This may be due to the overall poorer performance of Mi, resulting in more error trials which may result in stronger neural responses. Instead, Mo showed many cells in which we could decode information about the C-stimulus, including the gap direction, which is relevant to making a correct choice. Possibly, this was

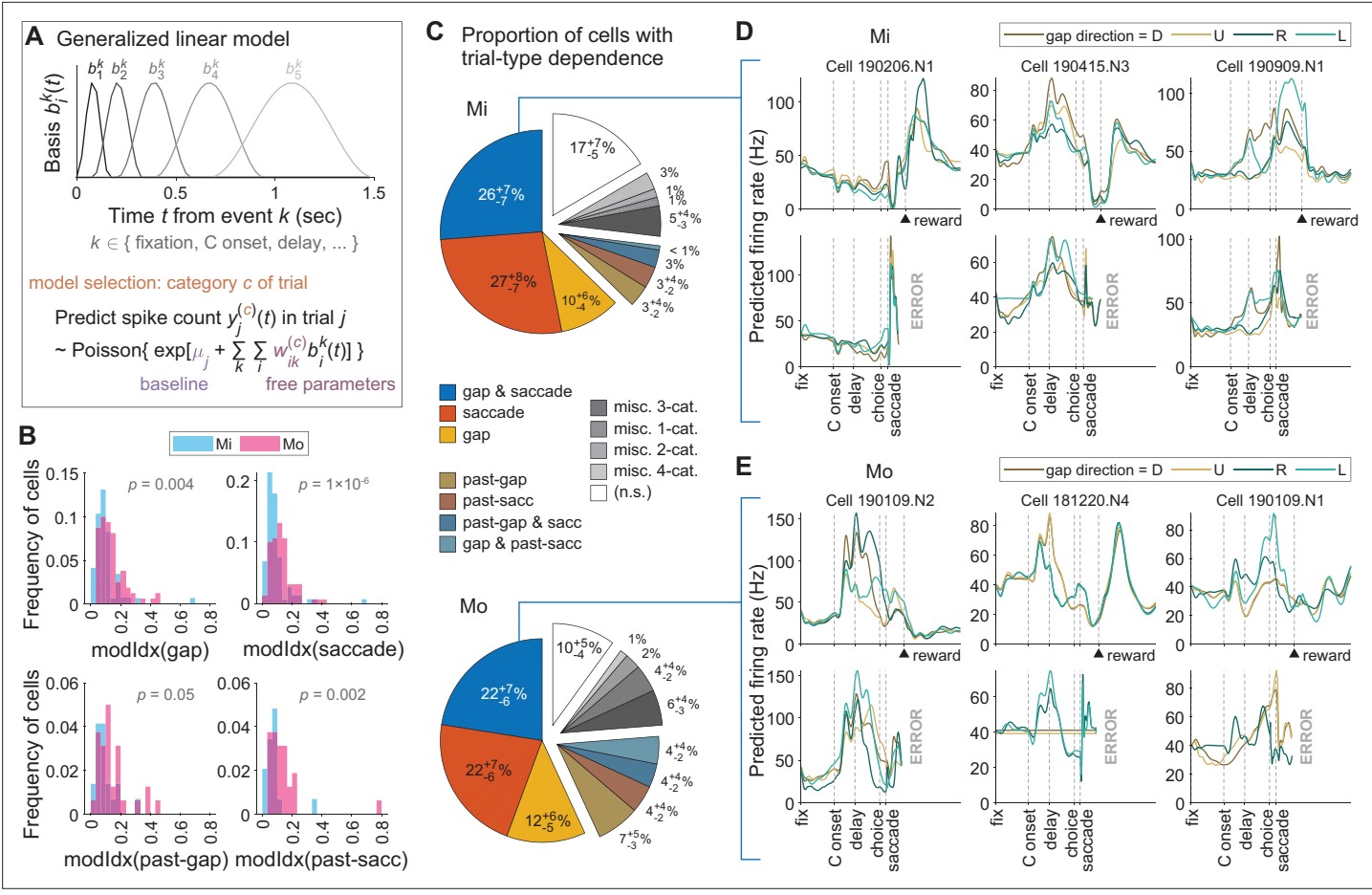

**Figure 3.** Encoding of task events by dentate nucleus (DN) neurons. (**A**) Schematic of a generalized linear model (GLM) for the spike count of a given neuron as being Poisson-distributed around behavior-dependent mean rates. The neuron is predicted to have time-varying changes in activity triggered by one or more behavioral events, and these changes are flexibly modeled as weighted sums of event-aligned basis functions $b_i^k(t)$. The set of events that the neuron responds to, as well as any trial-type specificity of these responses, was selected so as to obtain the most parsimonious model that produced a good fit. (**B**) Distribution across cells (two colors for the two monkeys) of a 'modulation index' (modIdx) score for how much trial types influence neural responses (see also examples for individual cells in panels D and E). First, we constructed an 'unspecialized' GLM model that predicts the cell's activity rate in trial $j$ to be $\lambda_j(t)$, independently of trial type. This is compared to a trial-type dependent GLM prediction $\lambda_j^{(c)}(t)$ for that cell, where the significant trial-type categories $c$ for the model were parsimoniously selected as described in the Methods. The modulation index for the cell is then defined using the ratio of trial-type specific to unspecialized model predictions: $\rho_j^{(c)}(t) \equiv \lambda_j^{(c)}(t)/\lambda_j(t) - 1$. To produce a single summary number per cell, we computed $modIdx^2$ as the time- and trial-type average of $\left[\rho_j^{(c)}(t)\right]^2$. The four subplots of this panel show the distribution of modIdx scores for cells with GLM predictions $\lambda_j^{(c)}(t)$ where the model-selected trial category $c$ included gap direction (top left), saccade direction (top right), past-trial gap direction (bottom left), and past-trial saccade direction (bottom right). (**C**) Proportions of cells with GLM models that had significant dependencies on various trial types as indicated in the legend of panel A. The miscellaneous 1- to 4-category proportions were pooled over cells with infrequently occurring trial-type dependencies (<2% cells per category); for example, the 'misc. 3-cat.' cells included those with simultaneous trial-type dependencies on gap, past-trial gap, and past-trial saccade conditions. (**D**) GLM prediction for the firing rate vs. time in a trial of three example neurons recorded in Monkey Mi. For each cell (one column), the predicted firing rate is shown for trials of four different gap directions (differently colored lines), and separately for correct trials (top row) and error trials (bottom row). The onset of various events in the trial are indicated with vertical dashed lines. (**E**) Same as D, but for three example cells from monkey Mo.

The online version of this article includes the following figure supplement(s) for figure 3:

**Figure supplement 1.** Examples of correct and incorrect trial and principal component analysis (PCA) decomposition of this population activity.

**Figure supplement 2.** Ramping activity of dentate nucleus (DN) neurons that modulated upon presentation of the C-stimulus.

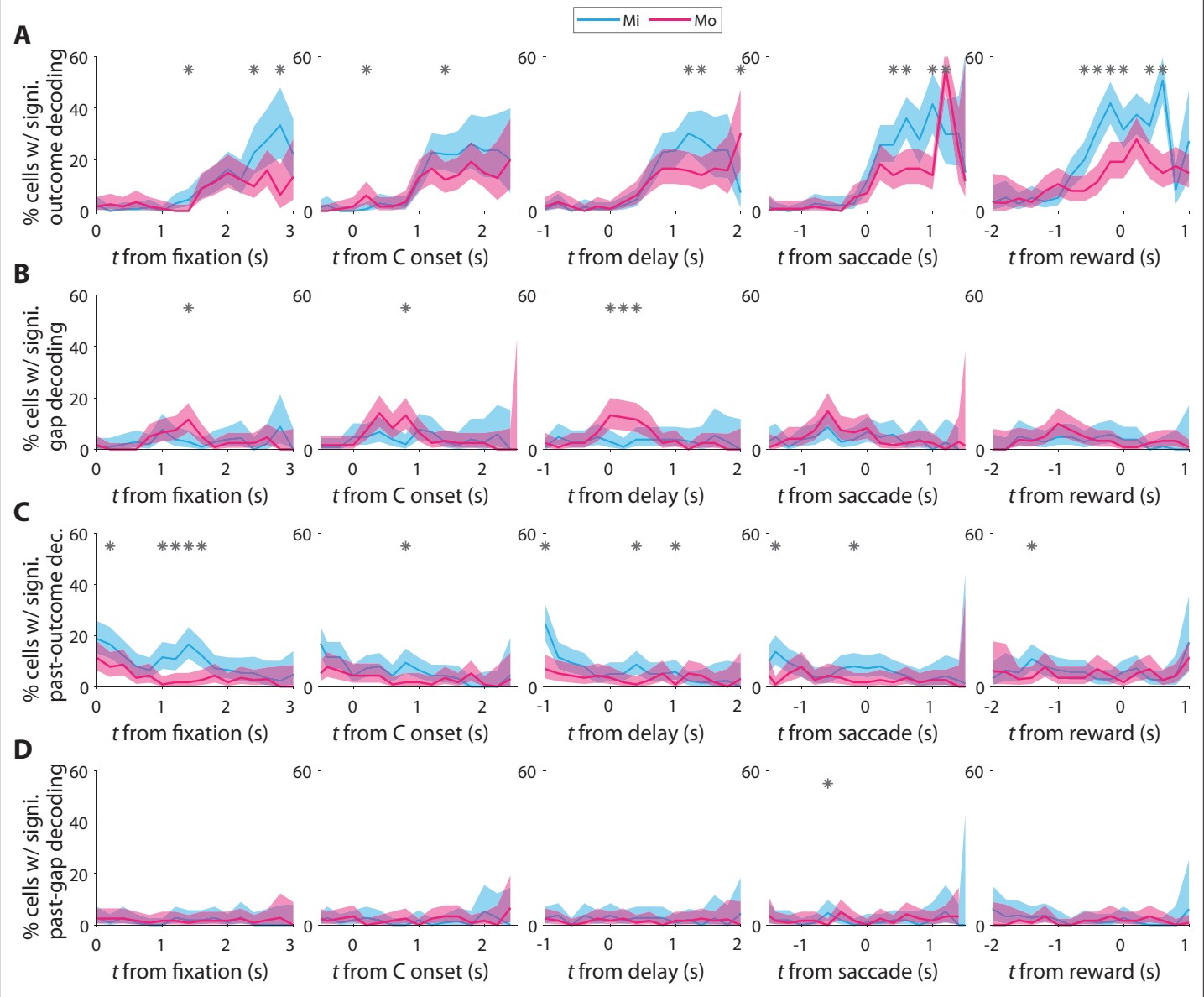

**Figure 4.** Decoding of trial conditions from neural firing rates. (**A**) Fraction of cells from which the trial outcome can be significantly decoded (see Methods) using data in 200-ms time bins aligned to various events in the trial (columns). * indicates bins in which this fraction is significantly different between the two monkeys (p < 0.05; Fisher's exact test, two-tailed). (**B–D**) Same as A but for the decoding of gap direction, outcome of the past trial, and gap direction of the past trial, respectively. Note the higher percentage of cells with decoding of the gap occurrence shortly after the delay period in Mo (**B**), and the higher percentage of cells with past-outcome decoding in Mi (**C**).

particularly visible in Mo as his DN activity was generally better entrained to the stimulus, leading to a better overall behavioral performance (*Figure 4B*). In line with the findings of the behavioral model, Mi instead had more cells than Mo that maintained information about the choice he had made in the previous trial, well into the time-frame of the ongoing trial. This information was notably present at about 1–1.5 s after fixation at the center, and gradually disappeared from the onset of the C-stimulus and beyond (*Figure 4C*). Lastly, there were almost no cells from which the gap direction of the past trial could be decoded in either NHP (*Figure 4D*).

## Identification of recording site in DN and its connectivity

The area of the DN with the most successful recordings was confirmed post mortem with the use of Cholera Toxin B (CTb) injections (*Figure 5A*). In both cases, the injections were focused on the

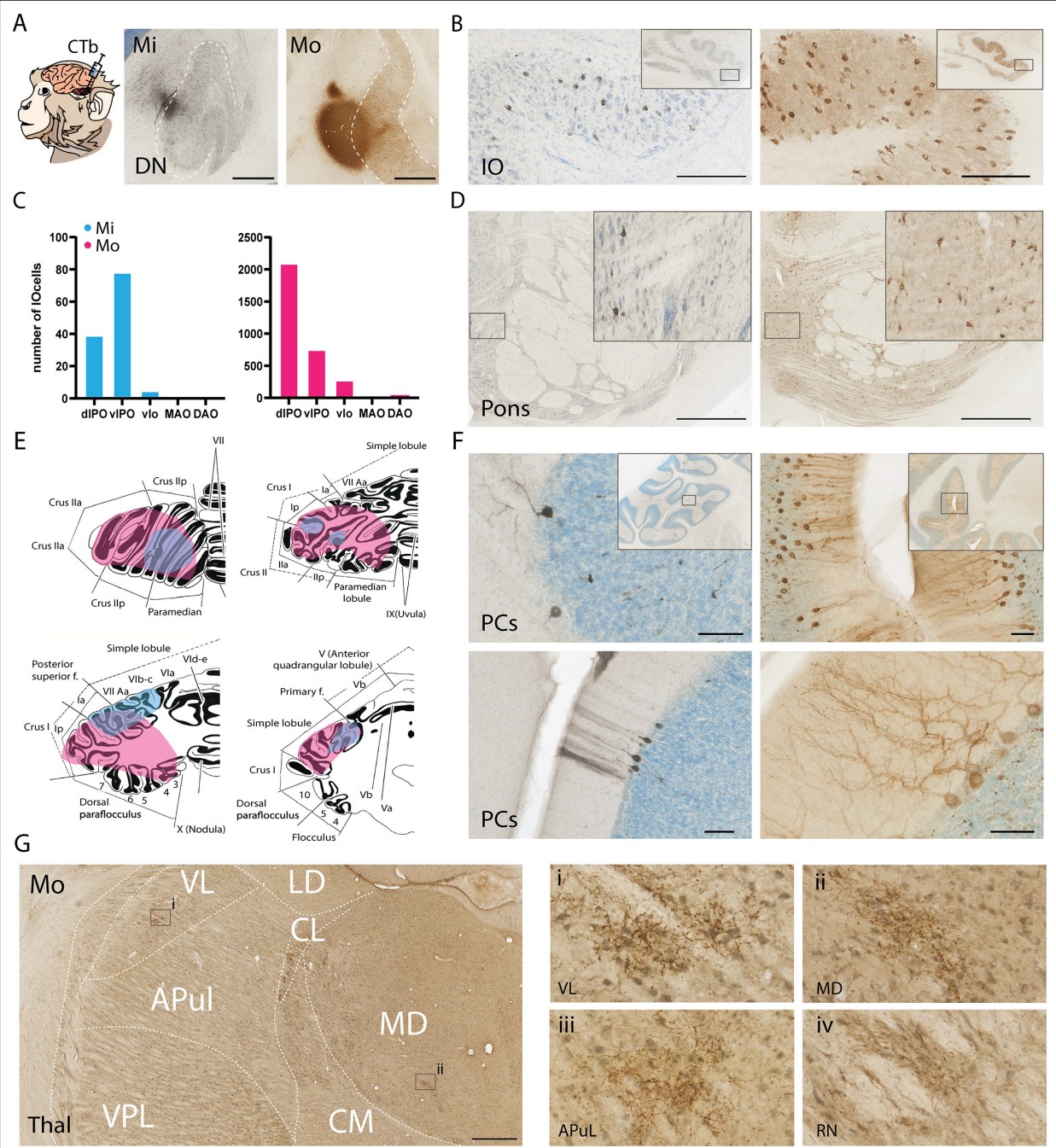

**Figure 5.** Anatomical characterization of recording area and its input and output connectivity. (**A**) Cholera Toxin B (CTb) injections in the dentate nucleus (DN) of Mi and Mo display the site of electrophysiological recordings of its neurons at the lateral edge of the DN. (**B**) Pictures of retrogradely labeled cells in the Principal Nucleus of the inferior olive (IO); insets show overview of IO. (**C**) Quantification of retrogradely labeled cells in different subnuclei of the IO in Mi (blue) and Mo (pink). (**D**) Examples of retrogradely labeled cells in the medial pons of Mi (left) and Mo (right); insets show higher magnification of labeled cells. (**E**) Schematic representations of cerebellar cortical areas containing retrogradely labeled Purkinje cells in Mi (left) and Mo (right). (**F**) Examples of retrogradely labeled Purkinje cells for both animals (Mi and Mo in panels on the left and right, respectively); insets show areas of the higher magnifications with the labeled cells. (**G**) Example of anterograde CTb labeling of cerebellar fibers with dense varicosities (arrowheads) in the thalamus of Mo. Panels i and ii on the right correspond to insets in the left panel; panels iii and iv are examples of anterograde CTb labeling outside of the left panel in the anterior pulvinar and red nucleus, respectively. VL, LD, CL, APul, VPL, CM, and MD indicate ventrolateral, laterodorsal, centrolateral, anterior pulvinar, ventroposterior lateral, centromedial, and mediodorsal nucleus of the thalamus, respectively. dlPO, vlPO: dorsolateral-, ventrolateral Principle Olive; vlo: ventrolateral outgrowth; MAO and DAO: medial and dorsal accessory olive. Scale bars in A, B, D, F, and G indicate 1250, 100, 1000, 50, and 1000 μm, respectively.

centrolateral part of the DN positioned in the center of its rostrocaudal axis; given the distribution of the modulating cells according to the stereotactic coordinates with respect to the 2D-grid of the recording chamber and the recording depths, we estimated that the functionally relevant recording area covered a vertical column of approximately 2 mm long, 2 mm wide, and 3 mm deep, as all modulated cells were recorded in a square of that size. Despite a similar location for the core of the injection site, the size of the injection was bigger in Mo as compared to Mi, possibly reflecting different stock preparations or anisotropic spread of the tracer. Accordingly, when we quantified the number of retrogradely labeled neurons in one out of four sections of the inferior olive (IO), so as to assess the relative effective size of the DN injections in both animals, we found more labeled cells in Mo (*Figure 5B, C*). In Mi the dorsolateral and ventrolateral leaf of the principal olivary nucleus (PO) contained 38 and 78 cells, respectively, while the ventrolateral outgrowth of the PO contained only 4 cells. Instead, in Mo we counted 1977, 763 and 312 neurons in the dorsolateral leaf, ventrolateral leaf, and ventrolateral outgrowth of the PO, respectively (*Figure 5C*). The labeling in the PO of the IO was in line with the retrograde labeling in the pons, which was predominantly confined to the medial pontine nuclei in both Mi and Mo, with Mo showing more labeled cells (*Figure 5D*). Likewise, Mo showed more retrogradely labeled Purkinje cells (PCs; *n* = 1314) than Mi (*n* = 84), when we screened 1 out of 24 sections of the cerebellar cortex (*Figure 5E, F*). Most of the labeled PCs appeared to be located in the D1 and D2 zones of the cerebellar cortex (*Voogd et al., 2012*).

Interestingly, the CTb injections into the DN also provided anterograde labeling of the climbing fibers, the distribution of which overlapped to a large extent with that of the retrograde labeling of the PCs in both Mi and Mo (lateral panels in *Figure 5F*). In addition, we found anterograde CTb labeling in the higher brainstem; this included for example the parvocellular and magnocellular red nucleus, zona incerta and deep mesencephalic nucleus as well as various thalamic nuclei, such as the ventro-lateral thalamic nucleus (VL), pulvinar nucleus (Pul), mediodorsal thalamic nucleus (MD), and centro-medial and centrolateral thalamic nuclei (*Figure 5G*). Instead, we did not find substantial labeling in the superior colliculus, ventral tegmental area or hypothalamus. The labeled areas we found are consistent with previous studies illustrating disynaptic pathways from the DN to cerebral cortical areas involved in visual processing such as the FEF area (*Kipping et al., 2013*; *Middleton and Strick, 2001*; *Romanski et al., 1997*).

## Discussion

We investigated to what extent the activity of DN cells in the cerebellum of NHPs encodes different aspects of a complex visual-motor Landolt C task. Choice performance was related to individual behavioral strategies of animals that changed over time during and after post-break relearning of the task. Accordingly, we found that DN cells connected with the D1 and D2 zones in the cerebellar cortex have highly diverse activity patterns. Their activity often bridges different experimental epochs and frequently show both upbound and downbound modulations (*De Zeeuw, 2021*), similar to PCs in the lateral cerebellum that project to the DN (*Avila et al., 2022*). Importantly, following training of the task, many DN neurons respond in a visual direction- or movement direction-selective manner more than half a second before the saccade of choice is made. Thus, multimodality and timing-dependent activation appear to be common properties of neurons in the lateral cerebellum.

### Visuomotor DN neurons connect to cerebellar modules in D1 and D2 zones

The distribution of the retrogradely labeled PCs in lateral lobules VII and VIII as well as crus I/II indicates that the region where we recorded in the DN forms part of the cerebellar D1 and D2 microcomplex (*De Zeeuw, 2021*; *De Zeeuw and Ten Brinke, 2015*). This conclusion is supported by the retrograde CTb labeling we found in the part of the medial pons that is known to receive direct input from the prefrontal cortex and to activate the D zones during decision making and motor planning (*Buckner et al., 2011*; *Prevosto et al., 2010*; *Ramnani, 2012*; *Salmi et al., 2010*; *Stoodley and Schmahmann, 2010*; *Strick et al., 2009*; *Wang et al., 2022*). Likewise, most of the retrograde labeling in the olive following the injections in the DN was present in the dlPO and vlPO, which are also part of the D1 and D2 modules (*Voogd et al., 2012*; *Voogd and Ruigrok, 2012*).

Even though both Mi and Mo showed retrograde CTb labeling patterns consistent with involvement of the D modules, the density of labeling differed substantially, presumably due to differential efficacy of the CTb injections (*Figure 5A*). Accordingly, we observed the same differential pattern when we analyzed the anterograde labeling results. Here too, Mo showed more densely labeled fibers in the various projection regions than Mi. Most anterogradely labeled fibers were found in the pulvinar, mediodorsal thalamus and ventrolateral thalamus, which are known to connect to various parts of the prefrontal cortex, such as FEF and Brodmann area 46 (*Kipping et al., 2013*; *Middleton and Strick, 2001*; *Romanski et al., 1997*), and/or the parietal eye field, an established hub in the dorsal attention network (*Andersen et al., 1992*; *Hanks et al., 2015*; *Roitman and Shadlen, 2002*; *Shadlen and Newsome, 2001*; *Svoboda and Li, 2018*). Interestingly, the dorsal attention network has been shown to connect to lobules VII and VIII (*Brissenden et al., 2016*; *Buckner et al., 2011*), precisely the location of the cerebellar cortex where retrograde labeling was detected following the DN injections. Taken together, the anatomical findings suggest that the DN region under study is at least partly integrated in a functional closed loop between cerebellum and cerebral cortex that appears to be involved in decision making, planning, and attention.

## Visuomotor processing by the DN

In the current study, the animals had to find a peripherally located C and identify the direction of a related gap, while fixating their eyes on a central fixation point. After a delay, the primates had to move their eyes to a virtual point in space dependent on the cue implied by the gap of the peripheral C-stimulus. The spatial and temporal dissociation of cue and action in this task allows us to separate neural activity related to the visual and attentional processes from that related to motor coordination (*Avila et al., 2022*; *Goldberg and Segraves, 1987*). Our data reveal that individual DN cells can encode both the direction of the visual cue and the associated direction of the subsequent saccade, suggesting that different task-specific activations in the cerebellar cortex may be relayed downstream to the same target neurons in the cerebellar nuclei. Our data are compatible with and expand upon fMRI studies, which have demonstrated that lateral lobules VIIb/VIIIa upstream of the DN exhibit functional properties characteristic of the dorsal attention network in the cerebral cortex (*Brissenden et al., 2018*; *Brissenden et al., 2016*; *van Es et al., 2019*). Indeed, this part of the cerebellar hemispheres shows task-specific activation in that their responses depend on representation of visuospatial location and/or load of working memory (*Brissenden et al., 2018*; *Brissenden et al., 2016*; *van Es et al., 2019*).

Our modeling and PCA of DN activity showed ramping activity in both primates, with better decoding of the gap direction and trial outcome, and shorter latencies following presentation of the visual stimulus in the better performing monkey (Mo). The difference in onset of DN ramping between the monkeys highlights the possibility that injecting accelerations of simple spike modulations of PCs in the cerebellar hemispheres into the complex of cerebellar nuclei may be instrumental in improving the performance of responses to covert attention, akin to what has been shown for the impact of PCs of the vestibulocerebellum on responses in the vestibular nuclei and compensatory eye movements upon vestibular stimulation (*De Zeeuw et al., 1995*). Likewise, our data on the differences in choice performance between Mo and Mi are also supported by the different reaction times in expert and amateur baseball players, who show cerebellar activation upon visual presentation of a pitched ball (*Owens et al., 2018*). Moreover, our recordings are corroborated by electrophysiological studies that have revealed a role of the cerebellar nuclei, including that of the DN, in motor planning (*Chabrol et al., 2019*; *Deverett et al., 2019*; *Deverett et al., 2018*; *Gao et al., 2018*). The DN cells recorded in the current study may contribute to the preparation of directed saccades through their projections to the FEF via the thalamus (*Lindeman et al., 2021*; *Middleton and Strick, 2001*; *Romanski et al., 1997*). Similar to our DN neurons, the FEF neurons that are embedded in the dorsal attention network encode multiple aspects of saccade planning and processing, including target identification in covert visual searches (*Brissenden et al., 2018*; *Brissenden et al., 2016*; *Buckner et al., 2011*; *van Es et al., 2019*), as well as decision making among multiple choices (*Ding and Gold, 2012*; *Gregoriou et al., 2012*; *Hanes and Schall, 1996*; *Monosov and Thompson, 2009*; *Schall and Hanes, 1993*). Accordingly, FEF neurons can show the type of ramping activity before upcoming eye movements (*Basu and Murthy, 2020*; *Raghavan and Joshua, 2017*) that we present here for the DN neurons. Given that such ramping activity is one of the main signatures of motor preparation (*Chabrol et al., 2019*;

*Deverett et al., 2019*; *Gao et al., 2018*; *Hanks et al., 2015*; *Svoboda and Li, 2018*), DN neurons may have a critical role in facilitating the preparation of visuomotor processing in general.

## Multimodal character of DN activity

The cerebellum receives inputs from many different external sensory as well as internal cognitive modalities, all of which can be integrated during the preparation and execution of complex tasks (*De Zeeuw, 2021*). In the cerebellar cortex such integration can take place at both the input and output stage. At the input stage, that is, at the level of the granule cells in the granular layer, activation of different sensory modalities has been shown to enhance spiking output (*Giovannucci et al., 2017*; *Ishikawa et al., 2015*). Moreover, given that many individual granule cells can receive inputs from both the cuneate nucleus relaying sensory proprioceptive signals and the pontine nuclei mediating presumptive motor command signals (*Guo et al., 2021*; *Huang et al., 2013*; *Wagner et al., 2019*), it is likely that sensory, motor and cognitive signals can also be integrated at the level of the cerebellar input layer. At the level of the PCs in the molecular layer with inputs from thousands of parallel fibers contacting single PC dendrites, the opportunities for convergence and integration of different modalities are even greater (*Gao et al., 2012*).

Our results demonstrate that integration of various forms of sensory, motor, and/or cognitive signals also occurs at the level of the DN neurons downstream of PCs in the D1 and D2 zones. Indeed, single DN neurons can encode the direction of a visual stimulus, as well as the preparation and execution of a saccadic eye movement into a particular direction. The encoding of both stimulus and saccade directions is present in the biggest proportion of task-modulated neurons in both monkeys. The decoding analysis showed similar outcomes, albeit with greater variability between Mi and Mo. To what extent the integration of different modalities is taking place at the level of the DN itself, mediated by converging inputs from the PCs, mossy fibers, and/or climbing fiber collaterals, or solely by events upstream in the cerebellar cortex, remains to be elucidated.

## Systems mechanisms underlying covert attention

Our electrophysiological data on DN encoding during the Landolt C task are in line with fMRI studies showing covert attention signals in the cerebellum (*Brissenden et al., 2018*; *Brissenden et al., 2016*; *van Es et al., 2019*). These neuroimaging studies, which have also implemented a consistent distractor stimulus to evaluate the validity of the task (*Posner, 1980*), point toward a coordination of cerebral and cerebellar cortical activation during covert attention. In this respect, both our electrophysiological study and the imaging studies of others elaborate on the premotor theory of covert attention, which has so far mainly highlighted the relevance of processing in the cerebral cortex (*Corbetta et al., 1998*; *Nobre et al., 2000*; *Rizzolatti et al., 1987*). More specifically, the premotor theory highlights the relevance of circuits in the frontal and parietal lobes that are activated during both attention shifts and subsequent movements, so as to strengthen the association process. Our data show that most of the neurons in the main output nucleus of the cerebellum, the DN, exhibit heterogeneous modulations during putative attention shifts made when the animals read the peripheral C-stimulus, as well as during the subsequent goal-directed saccadic eye movements that depend on this preceding cue in the peripheral visual field. The converging coordination of cerebellum and cerebral cortex in higher order mental processes such as covert attention also occurs during other cognitive processes. For example, not only the mirror neuron system in cerebral cortex but also connected areas in the cerebellum are actively involved in perceiving and interpreting the action of others (*Abdelgabar et al., 2019*). Indeed, just like during the Landolt C task (*Brissenden et al., 2018*; *Brissenden et al., 2016*; *van Es et al., 2019*), lobules VII and VIII in the cerebellar hemispheres are recruited during such action observation tasks when evaluating the kinematics of goal-directed hand actions of others (*Abdelgabar et al., 2019*). Moreover, patients suffering from spinocerebellar ataxia type 6 are severely impaired in performing these tasks (*Abdelgabar et al., 2019*). These data raise the question as to how the cerebellum may contribute to overt and covert attentional tasks (*Lupo et al., 2018*). One can hypothesize that cerebellar processing helps the frontal premotor and parietal cortices to map visual input from high level visual regions onto the motor machinery involved in performing related goal-directed actions. This suggests that much like action control (*Wolpert and Ghahramani, 2000*), complex mental observation, either overt or covert, relies on a cortico-cerebellar loop that maps sensory input onto motor control structures (inverse models) and motor programs to expected

sensory input (forward models). This loop may bring descending information from our visual cortical networks to the cerebellum and ascending information from the cerebellum back to premotor areas in the prefrontal cerebral cortex.

## Materials and methods

### Animals

All procedures complied with the NIH Guide for the Care and Use of Laboratory Animals (National Institutes of Health, Bethesda, Maryland), and were approved by the Institutional Animal Care and Use Committee of the Royal Netherlands Academy of Arts and Sciences (AVD8010020184587). The ARRIVE guidelines in line with the rules of the Netherlands Institute for Neuroscience were applied. Two adult male NHPs (*Macaca mulatta*), referred to as Mi and Mo, were used in this study. Before being subjected to the peripheral Landolt C saccade task of the current study, both of them were used for two other studies, one on glissade control and another one on mechanisms underlying anti-saccades (*Avila et al., 2022*; *Flierman et al., 2019*).

### Procedures for physiological experiments

#### Surgery

Animals were prepared for eye movement recordings as well as extracellular single-unit recordings in the cerebellum using a two-step surgical procedure (*Chen et al., 2017*). First, under general anesthesia induced with ketamine (15 mg/kg, i.m.) and maintained under intubation by ventilating a mixture of 70% $N_2O$ and 30% $O_2$, supplemented with 0.8% isoflurane, fentanyl (0.005 mg/kg, i.v.), and midazolam (0.5 mg/kg • hr, i.v.), we implanted a titanium head holder, which allowed for immobilization of the NHP's head. Four months later, under the same anesthesia conditions, a custom-made 40-mm-wide chamber was implanted to gain access to the cerebellum. Animals recovered for at least 21 days before behavioral training and testing (for more details on surgical procedures, see *Avila et al., 2022*; *Flierman et al., 2019*).

#### Behavioral experiments

While they were head-restrained, Mi and Mo were trained to focus on a fixation dot at the center of the monitor, which was placed at a viewing point distance of 52 cm and operating at a frame rate of 100 Hz (1152 × 864 pixels). Eye movements of their left eye were recorded with an infrared video eye tracker scanning at a 1000 Hz (Eyelink 1000 plus, SR Research) (*Figure 1*). Their eye movements were calibrated before every experiment (5 Standard Eyelink points for 10° eccentricity). We used the Landolt C task as described before (*Ignashchenkova et al., 2004*). At trial onset the animals were given a window of 500 ms to make a saccade to the central fixation point (red dot, 0.2° diameter) and required fixation to this point within a 4° diameter circular window for a period of at least 100 ms. If fixation was broken, a new trial was started after a 1.5-s time-out. If not, the red C-stimulus was presented in one out of four positions (*Figure 1C*), randomly selected every trail. The gap direction of the C was always perpendicular to the position of the C relative to the fixation dot. The C-stimulus was presented for 250 ms at 5° eccentricity and had an outer and inner diameter of 1° and 0.6°, respectively. During the presentation of the C-stimulus the animals had to maintain fixation on the fixation dot. After the 250-ms period, the C-stimulus was masked for 100 ms with a full red circle to prevent the animals from gaining extra information from the retinal after-images. After the offset of the mask presentation, the fixation dot was visible for another randomly drawn interval between 500 and 700 ms (50 ms steps) during which the animals were also not allowed to break fixation. After this delay period, two potential saccade targets were presented perpendicular from the target location; subsequently, the fixation dot turned gray (go-cue), indicating that the animal was allowed to make a saccade into the direction indicated by the initial cue, that is, the gap of the C-stimulus. A trial was rewarded with a juice reward when the animals made a saccade to the correct target indicated by the gap of the C-stimulus within 500 ms after the central target turning gray. The correct target had to be fixated within a 4° diameter circular window around the target and fixation had to last at least 100 ms. Incorrect trials, in which the animal made a saccade to the wrong target, or if any of the before mentioned requirements were not met, triggered a 1.5-s time-out and the reward duration was reset to a minimum value of 100 ms (see next section for details on juice reward).

## Motivation and controlled water intake

Animals had restricted water intake on the day before the experiment and they were water deprived on the day of the experiment. Controlled water intake was performed in consultation with the animal caretakers and in accordance with the rules of the Dutch law. During the task the animals were allowed to drink as much as they liked, yet always in the context of a reward for correctly executed trials. Rewards were in the form of strawberry or tropical lemonade, depending on the animal's preference. The duration and thereby amount of reward delivery started with a pulse of 100 ms. To motivate the animal to perform well, 50 ms was added to the reward pulse for every consecutively correct trial, up to a maximum 350 ms. If the animal made an incorrect response, no reward was given for that trail and the duration of the reward was reset to the initial 100 ms. After the experiment the animals were given water ad libitum.

## Electrophysiological recordings

MRI images were used as a reference frame for anatomical localization of the left DN (*Figure 2A*). Single-unit recordings were obtained using tungsten glass-coated electrodes (1.5 MΩ, Alpha Omega Engineering, Nazareth, Israel) through a 23-gauge guide tube, which was inserted only through the dura. A motorized microdriver (Alpha Omega Engineering, Nazareth, Israel) with a 1-mm spaced grid was used to introduce the electrode and map the recording sites. Extracellular recordings were digitized and sampled at 44 kHz and subsequently stored during the experiment using a Multi-Channel Processor (Alpha Omega Engineering, Nazareth, Israel). Single units were determined to be DN neurons by the $x$, $y$, and $z$ coordinates of the electrode tip in relation to those of the MRI images, as well as their waveform in combination with the absence of complex spikes. The discovery of a task-related site was often guided by a facilitation or suppression of single-unit action potentials in relation to task-related variables, such as the C-stimulus, saccade, and/or reward. The electrophysiological data, as detected with an online spike sorter (Multi-Spike Detector, Alpha Omega Engineering), were stored for offline analysis using custom-written MATLAB code.

## Analyses of behavioral and electrophysiological data

### Eye movement analysis

Saccade onset and offset were detected on the basis of an adaptive velocity threshold, which consisted of 3 standard deviations (std) of the noise during fixation (see also *Flierman et al., 2019*). A Savitzky–Golay filter was applied for smoothing the raw traces. Position traces were differentiated to find the eye velocity and acceleration signals.

### Neuronal modulation analysis

Task-related neuronal modulation was determined by the application of 200-ms sliding windows around the main events of interest (i.e., C-stimulus onset, saccade onset) (*Table 1*). Spike counts in the windows were compared with baseline activity (taken during −500 to 0 ms from fixation). If average activity over different trials in one of the sliding windows was significantly different from the average activity in the baseline window, the activity of that neuron was considered to be significantly modulated for that event of interest. In case of the stimulus event, if more than half of the windows were significantly different from the baseline window, the cell was considered to have sustained activity during the epoch surrounding the stimulus event.

### Parametric statistics

If statistical testing involved two groups, and the data were normally distributed, a Student's *t*-test was used. When more than two groups were involved, we used ANOVA tests with the Tukey–Kramer post hoc test. To determine directional selectivity of neurons we used one-way ANOVA unless specified

**Table 1.** Description of the sliding windows used to determine significant modulation for each category.

| Event | Description |
| --- | --- |
| Stimulus | 0–800 ms after C-onset; 200 ms sliding window with 100 ms steps, centered around 100–700 ms |
| Saccade | From 200 ms before until 200 ms after saccade onset; 200 ms sliding window with 100 ms steps |

otherwise. When comparing fractions, the Chi$^2$ for proportions was applied to determine significance. For circular statistics, the circle statistics toolbox in MATLAB was used.

## Directional selectivity

For each of the four possible stimulus directions (i.e., C-stimulus position, or gap direction, dependent on sorting trials) we determined the time point in the stimulus window and the saccade window where the difference between the minimum and maximum spike rate was biggest. In a 200-ms window around that time point, the spike rate per trial was determined. ANOVA was computed for the spike rate in the window with the different directions as a grouping variable. If the test was significant, the modulation was determined to be direction selective ($\alpha$ = 0.05). The direction that showed the biggest change in firing rate from the baseline was selected as the preferred direction. To determine if a cell preferentially modulated for the stimulus or action direction, that is, the preferred direction sorting, the ANOVA with the lowest p-value was selected. Given that the stimulus direction and action direction are always mutually perpendicular in our task design (*Figure 1A*), these two parameters are inherently correlated.

## Behavioral model

To determine which task modality and behavioral variables (e.g., gap direction and saccade direction in trial $n - 1$) were most predictive of outcome response at the end of a trial (e.g., left or rightward saccade in trial *n*) a logistic regression model was fitted. Because logistic regression models data with binary outcomes, the model was fitted separately for the L/R gap direction trials (i.e., L/R task) and U/D gap direction trials (i.e., U/D task; *Figure 1A*). This logistic regression model provides a description of the data in terms of weights of behavioral variables that best predict the outcome of the task. Confidence intervals were created by performing 100 bootstrap permutations and represent 5–95% percentiles.

## Encoding model

To more fully understand the dynamics of neural responses over the time course of a trial, we fitted a GLM to predict the spike train data of each neuron as temporally overlapping responses to behavioral events in the trial, using the method and code described by *Park et al., 2014*. In brief, the GLM predicts the probability $P\left[y_j\left(t\right) \mid \vec{\tau}_j, \vec{w}\right]$ of a given neuron producing a number $y_j\left(t\right)$ of spikes in a 25-ms time window at time $t$ in trial $j$. This probability is a function of the times $\vec{\tau}_j$ of various behavioral events in that trial, and has free parameters $\vec{w}$ that were fit to data as described below. The model assumes that the neuron's spike counts are Poisson distributed around the mean spike rate $\lambda_j\left(t\right) = exp\left[\mu_j + \sum_k\sum_i w_{ik}b_i^k\left(t\right)\right]$, where $\mu_j$ is the baseline firing rate of the neuron in trial $j$ (moving average firing rate using a 5-trial window). $b_i^k\left(t\right) \equiv b_i\left(t - \tau_k\right)$ are linear kernels (basis functions) that capture the time-varying response of the neuron to the behavioral event at time $\tau_k$. We chose $b_i\left(\tau\right)$ (the basis functions aligned to the start of the event) to be a series of five ($i \in \{1, \ldots, 5\}$) cosine-shaped bumps with progressively larger widths and spanning up to 1.5 s from the start of the event. The weights $w_{ik}$ for each of these basis functions allow the neuron's modeled response to have a flexible time course, including the possibilities of rising above baseline (facilitation) if $w_{ik} > 0$, falling below baseline (suppression) if $w_{ik} < 0$, or even switching in time between the two, depending on the relative values of $\vec{w}_k \equiv \{w_{ik} \mid i = 1, \ldots, 5\}$. Six behavioral events were included in the model: the start of fixation (beginning of the trial), onset of the C-stimulus, start of the delay period, time of appearance of the choice target, time at which the saccade begins, and (for correct trials only) the time at which the reward is delivered (see *Figure 1B*). To account for the possibility that the neuron responds to only a subset of behavioral events, we fitted the model using sparse group LASSO regularization penalty (*Huang and Zhang, 2010*) with software from *Mairal et al., 2014*. The variable groups in the penalty terms correspond to $\vec{w}_k$, that is, all the basis-function weights for the response to a given behavioral event $k$, so that the fitted model will have $\vec{w}_k = \vec{0}$ (all weights being exactly zero) if a dependency on event $k$ does not significantly improve the model prediction. We selected the regularization strength by maximizing the fivefold cross-validated model likelihood.

Because neural responses also differed across subsets of trials with different task conditions (e.g., the orientation of the C-stimulus gap direction), we additionally fitted a variant of the GLM where the

model parameters depended on the category $c$ (defined below) of the trial. This model thus predicts that the neuron has spike counts $P\left[y_j^{(c)}(t) | \vec{\tau}_j, \vec{w}^{(c)}\right]$ in trials of category $c$, with potentially different parameters $\vec{w}^{(c)}$ for every category $c$. We call the previous model with no category dependence the 'unspecialized' model, and the model with category dependence the 'category-modulated' model. We considered all possible categories of trials as specified by four types of trial conditions: C direction, gap direction, saccade direction, and values of the same in the previous trial (past-gap and past-saccade). The gap direction (same for saccade direction) can take on one of four values {U,D,L,R}, which means that there are 14 different ways to construct categories of trials by gap direction (all partitions of the set). These 14 possibilities are:

|    | Categories |
|----|------------|
| 1  | {U}, {D}, {L}, {R} |
| 2  | {U, D}, {L, R} |
| 3  | {U, L}, {D, R} |
| 4  | {U, R}, {D, L} |
| 5  | {U}, {D, L, R} |
| 6  | {D}, {U, L, R} |
| 7  | {L}, {U, D, R} |
| 8  | {R}, {U, D, L} |
| 9  | {U}, {D}, {L, R} |
| 10 | {U}, {L}, {D, R} |
| 11 | {U}, {R}, {D, L} |
| 12 | {D}, {L}, {U, R} |
| 13 | {D}, {R}, {U, L} |
| 14 | {L}, {R}, {U, D} |

We furthermore wished to include the possibility that neuronal responses depend simultaneously on two or more trial conditions, for example, both the current trial gap and past-gap directions. As the number of all possible combinations of all trial-condition categories is exceedingly large, we used a greedy model selection procedure to select a parsimonious set of trial conditions that describes a given neuron's response sufficiently well. Starting from the unspecialized model as the 'parent', we first constructed a set of candidate 'child' models, each of which corresponds to a particular choice of categorization for a particular trial condition (e.g., one of these would be {U,D},{L},{R} for saccade direction). We then selected the child model with the highest (cross-validated) likelihood $L$ relative to its parent model, and if this is sufficiently high ($L_{child} > 10 L_{parent}$), we retain this child model as the best model so far. This best model was then used as the parent model for a second repetition of the same procedure. In subsequent repetitions, trial conditions that were already included in the parent model were no longer considered for the child model (e.g., if the parent model had saccade-direction categories, then its child models will only include additional dependencies on gap, past-gap, and past-saccade categories). This procedure terminates when adding categories to the model no longer significantly improves the model likelihood.

## Principal component decomposition

The average firing rate for correct and incorrect trials was calculated for the population of 145 and 160 neurons for Mi and Mo, respectively. From the firing rates in the window of [−0.5 to 1.5] s around stimulus onset PCA was performed for the pool of neurons of both monkeys, using the 'pca' function in MATLAB. Principal component coefficients were plotted over time to display new axes representing the direction of maximum variation on the trial average firing rates of correct and incorrect trials of individuals.

## Detection of firing rate ramping onset

After the PCA analysis, we further investigated whether differences in the latencies of neuronal ramping activity following presentation of the visual stimulus may contribute to heterogeneity in performance. To determine the latency of onset of both facilitating and suppressing firing rate ramping after presentation of the C-stimulus, a modified Kneedle algorithm was applied (*Satopaa et al., 2011*). This algorithm calculates the distance between each data point and the line connecting the starting point and the maximum point; the point with the greatest distance was identified as the ramping onset. To minimize noise from firing frequency fluctuations, average firing rates for correct and incorrect trials were generated using a 100-ms Gaussian kernel density estimation. Furthermore, due to the sensitivity of Kneedle for noise, the analysis was limited to neurons from which at least 100 correct and incorrect trials were recorded and that were found to be modulated during the 'stimulus window' as described above. Following the feature extraction process described above, statistical tests were conducted to assess significance through repeated measures ANOVA with the Tukey post hoc test. Latency until ramping onset was the outcome variable, trial outcome (i.e., correct or incorrect) as within subject variable, and monkey (Mi or Mo) as between subject variable.

## Decoding model

We used logistic regression to decode four behavioral quantities from the spiking activity of each neuron as a function of time in the trial (separately for each 200-ms time-bin aligned to different behavioral events). These behavioral quantities are the C direction, gap direction, the outcome of the trial, that is, whether the monkey receives a reward at the end of the current trial, and the previous trial versions of these quantities (past-gap direction and past-outcome). Because the trial outcome (e.g., variable A) and gap directions (variable B) are both behaviorally correlated, neural information about one behavioral quantity (neural activity being correlated with A) can spuriously appear as if it was information about another (neural activity being correlated with B, but only through the *behavioral* correlation between A and B). To control for this issue we used a trial weighting procedure to 're-balance' the dataset so that all pairs of behavioral variables are in effect uncorrelated (*Kobak et al., 2016*). For a given neuron, we first divided the trials in its dataset into categories defined by the three other behavioral quantities of interest (see above). Since the trial outcome can take on two values (rewarded/not rewarded), and gap direction can take on four values (U/D/L/R), this means that there are 2 x 4 x 2 x 4 = 64 categories of trials. We then computed the weight $w_i^c$ of trial $i$ in category $c$ as $w_i^c = 1/n^c$, where $n^c$ is the total number of trials in category $c$.

## Neuroanatomical tracing

### Injections

At the end of the experiments, 18–19 days before perfusion, the recording areas of the DN of the animals were injected with 600 nl 0.8% of the tracer CTb (Sigma-Aldrich, the Netherlands) dissolved in 0.01 M phosphate-buffered saline. Injection position was verified with an electrode that was fixed to the injection line. When performing the tracer injections the tip of the electrode exceeded the tip of the capillary by 1 mm, the positional difference was corrected for by descending this distance deeper. The injectrode was lowered through the tentorium in an extra wide guide tube, allowing simultaneous access of both the electrode and the capillary. After perforation of the tentorium the tissue was left to settle for 20 min so as to allow the wider guide tube and injectrode to acquire a stable position. Next, the injectrode was lowered another 5–8 mm over 30 min so that the tissue was as stable as possible during the injection. Guided by the earlier neuronal recordings and current activity on the electrode, the tip of the capillary was lowered to a depth that was estimated to be the center of the task-related site. The 600 nl of the CTb solution was injected over a time course of 10 min, after which the injectrode was left in place for another 15 min. Likewise, the injectrode was retracted into the guide tube over the course of 15–20 min.

### Perfusion

NHPs were transcardially perfused. The animals were deeply anesthetized with fentanyl and pentobarbital in their home cages, after which they were transported to the perfusion room. To prevent blood coagulation 2000 IE/kg of heparin was injected i.v. with 50 ml saline 15 min before the first incision was

made. First, the cardiovascular system was flushed with 2–3 l of 0.9% NaCl (saline) on a high perfusion speed of ~120 RPM, until the blood ran clear. Subsequently, the body was flushed with 3 l fixative of 4% paraformaldehyde in 0.1 M phosphate buffer (pH 7.6) in saline (PBS) with speed of the perfusion adjusted to 70 RPM, followed by 1.5 l of 4% paraformaldehyde with 4% sucrose in 0.1 M PBS (pH 7.6, at 70 RPM). Thereafter, the brain was dissected and post-fixed in 4% paraformaldehyde for up to 1 week.

### Immunocytochemistry

After post-fixation, brains were submerged in 10% sucrose PBS solution at 4°C until they sank. The brains were then embedded in 14% gelatin and 10% sucrose, after which they were fixated in 10% formalin and 30% sucrose overnight, and subsequently in 30% sucrose until they sank. The brains were then cut in 50 µm sections. For light-microscopic staining of the CTb, the sections were first incubated for 20 min in 3% $H_2O_2$ (in PBS) to remove endogenous peroxidase activity of blood. Next, the sections were blocked in 10% normal horse serum (NHS) in PBS with 0.5% Triton, after which they were incubated at room temperature with anti-CTb (goat, 1:15,000, List labs) primary antibody (in PBS with 2% NHS, 0.5% Triton) for 48–72 hr and next with biotinylated rabbit anti-goat (1:200, Vector) secondary antibody (also in PBS with 2% NHS and 0.5% Triton) for 1.5 hr. After washing with PBS, the sections were incubated in the Avidin–Biotine complex (1:200 Avidine and Biotine, Vector) in PBS and 0.5% Triton for 1.5 hr at room temperature. After washing with 0.5 M PB sections were incubated for several minutes with cobalt acetate 0.5% in MilliQ or directly washed with MilliQ, and subsequently briefly incubated with DAB (33.3 mg/100 ml 3,3'-diaminobenzidine; Sigma) and 17 µl $H_2O_2$ for visualization of the CTb. After washing with 0.5 M PB, all sections were mounted with chrome alum and counterstained with thionin.

### Image analysis

Sections were inspected under a Leica (Nussloch, Germany) DM-RB microscope, and relevant sections were identified for further scanning with a Hamamatsu Nanozoomer 2 whole slide imager. The resulting images were then exported as TIFF format for final quantification using Neurolucida software (MicroBrightField, Inc, Colchester, VT). Sections of the cerebellar cortex (1 in 12), inferior olive (1 in 6) as well as pons (1 in 6) were imaged at 20× for quantification of retrogradely labeled neurons. For identification of the injection site, a 1 in 6 series of the cerebellar nuclei was inspected. For histological delineations of neuroanatomical structures of the monkey brains we used three different atlases (*Gilman, 1972*; *Paxinos et al., 2000*; *Gilman, 1972*), adhering to the most common nomenclature. Sections of the mesencephalon and diencephalon were imaged with the nanozoomer (20× objective) and analyzed with NDP-view (Hamamatsu) software.

## Acknowledgements

Financial support was provided by the Netherlands Organization for Scientific Research (NWO-ALW 824.02.001; CIDZ), the Dutch Organization for Medical Sciences (ZonMW 91120067; CIDZ and Vidi/ZonMw/917.18.380,2018; AB), Medical Neuro-Delta (MD 01092019-31082023; CIDZ), INTENSE LSH-NWO (TTW/00798883; CIDZ), ERC-adv (GA-294775 CIDZ), and ERC-POC (nrs. 737619 and 768914; CIDZ); The NIN Vriendenfonds for Albinism the Dutch NWO Gravitation Program (DBI2; CIDZ) and Erasmus MC Convergence Flagship (Integrative Neuromedicine Convergence Health and Technology 2022; AB). We thank Kor Brandsma and Anneke Ditewig for biotechnical assistance and animal care-taking; Beerend Winkelman and Si-yang Yu for fruitful discussions and suggestions for analyses; Masaki Tanaka and Jun Kunimatsu for sharing the design of the injectrode used for the tracer injections; and Erika Goedknecht for her indispensable contributions to the processes of anatomical tracing and perfusion. Finally, we would like to thank Peter Strick for his advice on the anatomical tracing part of the study.

# Additional information

## Funding

| Funder | Grant reference number | Author |
| --- | --- | --- |
| Netherlands Organisation for Scientific Research | NWO-ALW 824.02.001 | Chris I De Zeeuw |
| ZonMw | ZonMW 91120067 | Chris I De Zeeuw |
| Medical Neuro-Delta | MD 01092019-31082023 | Chris I De Zeeuw |
| INTENSE LSH-NWO | TTW/00798883 | Chris I De Zeeuw |
| European Research Council | GA-294775 | Chris I De Zeeuw |
| European Research Council | 737619 and 768914 | Chris I De Zeeuw |
| NIN Vriendenfonds for Albinism | | Chris I De Zeeuw |
| Netherlands Organisation for Scientific Research | Dutch NWO Gravitation Program DBI2 | Chris I De Zeeuw |
| Netherlands Organisation for Scientific Research | Vidi/ZonMw/917.18.380,2018 | Aleksandra Badura |
| Erasmus Medical Center | Convergence Flagship | Aleksandra Badura |
| Howard Hughes Medical Institute | | Sue Ann Koay |
| European Research Council | Advanced Grant 101052963 'NUMEROUS' PR | Pieter Roelfsema |

The funders had no role in study design, data collection and interpretation, or the decision to submit the work for publication.

## Author contributions

Nico A Flierman, Conceptualization, Data curation, Software, Formal analysis, Validation, Investigation, Visualization, Methodology, Writing – original draft, Writing – review and editing; Sue Ann Koay, Software, Formal analysis, Writing – original draft, Writing – review and editing; Willem S van Hoogstraten, Resources, Data curation, Formal analysis, Validation, Visualization, Writing – original draft; Tom JH Ruigrok, Resources, Data curation, Validation, Visualization; Pieter Roelfsema, Conceptualization, Resources, Supervision; Aleksandra Badura, Conceptualization, Data curation, Supervision, Funding acquisition, Investigation, Methodology, Writing – original draft, Project administration, Writing – review and editing; Chris I De Zeeuw, Conceptualization, Supervision, Funding acquisition, Investigation, Methodology, Writing – original draft, Writing – review and editing

## Author ORCIDs

Nico A Flierman ⓘ https://orcid.org/0000-0003-3911-8228
Sue Ann Koay ⓘ https://orcid.org/0000-0002-9648-2475
Willem S van Hoogstraten ⓘ https://orcid.org/0000-0002-3635-4893
Tom JH Ruigrok ⓘ https://orcid.org/0000-0001-5537-1165
Pieter Roelfsema ⓘ https://orcid.org/0000-0002-1625-0034
Aleksandra Badura ⓘ https://orcid.org/0000-0002-0119-5108
Chris I De Zeeuw ⓘ https://orcid.org/0000-0001-5628-8187

## Ethics

All procedures complied with the NIH Guide for the Care and Use of Laboratory Animals (National Institutes of Health, Bethesda, Maryland), and were approved by the Institutional Animal Care and Use Committee of the Royal Netherlands Academy of Arts and Sciences (AVD8010020184587). Two adult male non-human primates (NHPs, Macaca mulatta) were used in this study.

Reviewer #1 (Public review): https://doi.org/10.7554/eLife.99696.3.sa1
Reviewer #2 (Public review): https://doi.org/10.7554/eLife.99696.3.sa2
Author response https://doi.org/10.7554/eLife.99696.3.sa3

## Additional files

### Supplementary files
Supplementary file 1. Regression analyses applied to the data from *Figure 1—figure supplement 1*. Only two of the regression models do significantly better than a constant model (F-test, highlighted in bold). Both of those do not survive a Bonferroni correction ($\alpha = 0.0083$ for 0.05/6 test per monkey).

Supplementary file 2. Descriptive statistics of latency to ramping onset.

MDAR checklist

### 12341Data availability
Electrophysiology data is available through on Dryad (https://doi.org/10.5061/dryad.1rn8pk14s). Associated MATLAB code for analysis is available on GitHub (https://github.com/sakoay/ErasmusAttn_2020; copy archived at *Koay, 2024*).

The following dataset was generated:

| Author(s) | Year | Dataset title | Dataset URL | Database and Identifier |
|---|---|---|---|---|
| Flierman NA | 2025 | Cerebellar Dentate Neurons | https://doi.org/10.5061/dryad.1rn8pk14s | Dryad Digital Repository, 10.5061/dryad.1rn8pk14s |

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
