## [Editor Report · eLife Assessment]

This **important** study examined neuronal activity in the dentate nucleus of the cerebellum when monkeys performed a difficult perceptual decision-making task. The authors provide **convincing** evidence that the cerebellum represents sensory, motor, and behavioral outcome signals that are sent to the attentional system. This paper is of great general interest in that it shows the involvement of the cerebellum in cognitive processes at the neuronal level.

---

## [Referee Report · Reviewer #1 (Public review)]

Summary:

Recordings were made from the dentate nucleus of two monkeys during a decision-making task. Correlates of stimulus position and stimulus information were found to varying degrees in the neuronal activities.

Strengths:

A difficult decision-making task was examined in two monkeys.

Weaknesses:

One of the monkeys had difficulty learning the task. The initial version of the manuscript lacked a coherent hypothesis to be tested, although the revision has improved things. In its current form, the manuscript does not provide data regarding the possibility that this part of the brain may have little to do with the task that was being studied. As noted in the response to the reviewer's comments, future studies could address this issue by providing results of additional inactivation experiments.

---

## [Referee Report · Reviewer #2 (Public review)]

The authors trained monkeys to discriminate peripheral visual cues and associate them with planning future saccades of an indicated direction. At the same time, the authors recorded single-unit neural activity in the cerebellar dentate nucleus. They demonstrated that substantial fractions of DN cells exhibited sustained modulation of spike rates spanning task epochs and carrying information about stimulus, response, and trial outcome. Finally, tracer injections demonstrated this region of the DN projects to a large number of targets including several known to interconnect the visual attention network. The data compellingly demonstrate the authors' central claims, and the analyses are well-suited to support the conclusions. Importantly, the study demonstrates that DN cells convey many motor and nonmotor variables related to task execution, event sequencing, visual attention, and arguably decision-making/working memory.

---

## [Author Response]

The following is the authors’ response to the original reviews.

**Reviewer #1:**
- Summary:Recordings were made from the dentate nucleus of two monkeys during a decision-making task. Correlates of stimulus position and stimulus information were found to varying degrees in the neuronal activities.

We agree with this summary.

- Strengths:A difficult decision-making task was examined in two monkeys.

We agree with this statement.

- Weaknesses:One of the monkeys did not fully learn the task. The manuscript lacked a coherent hypothesis to be tested, and no attempt was made to consider the possibility that this part of the brain may have little to do with the task that was being studied.

We understand the reviewers concern. It is correct that one of the monkeys (Mi) did not perform at a high level, but it should be noted that both monkeys learned significantly above chance level. Therefore, we would argue that both monkeys in fact did learn the task but Mi’s performance was suboptimal. This difference in the performance levels gave us a rare opportunity to dive deeper into the reasons why some animals perform better than the others and we show that Mi (the lower performing monkey) paid more attention to the outcome of the previous trial – this is evident from our behavioural and decoding models.

We tested the overall hypothesis that neurons of the nucleus dentate can dynamically modulate their activity during a visual attention task, comprising not only sensorimotor but also cognitive attentional components. Many neurons in the dentate are multimodal (Figure 3C-D) which was something that was theorized. One of the specific hypotheses that we tested is that the dentate cells can be direction-selective for both the sensorimotor and cognitive component. Given that many of the recorded cells showed direction-selectivity in their firing rate modulation for gap directions and/or stimulus directions, we provide strong evidence that this hypothesis is correct. We have now spelled out this hypothesis more explicitly in the introduction of the revised version. We now also explain better why we tested this specific hypothesis. Indeed, earlier studies in primates such as those by Herzfeld and colleagues (2018, Nat. Neuro.) and van Es and colleagues (2019, Current Biol) have indicated that direction-selectivity of cerebellar activity may occur in various sensorimotor domains.

We also appreciate the comment of this Reviewer that in our original submission we did not show our attempt to consider the possibility that this part of the brain may have little to do with the task that was being studied. We in fact did consider this possibility in that we successfully injected 3 ml of muscimol (5 μg/ml, Sigma Aldrich) into the dentate nucleus in vivo in one of the monkeys (Mo). This application resulted in a reduction of more than 10% in correct responses of the covert attention task after 45 minutes, whereas the performance remained the same following saline injections. Unfortunately, due to the timing of the experiments and Covid19-related laboratory restrictions we were unable to perform these experiments in the other monkey or repeat them in Mo. We aim to replicate this in future experiments and publish it when we have full datasets of at least two monkeys available. For this paper we have prioritized our tracing experiments, highlighting the connections of the dentate nucleus with attention related areas in brainstem and cortex in both monkeys, following perfusion.

- Perhaps the large differences in performance between the two subjects can be used as a way to interpret the neural data's relationship to behavior, as it provided a source of variance. This is what we would hypothesize if we believed that this area of the brain is playing a significant role in the task. If one animal learns much more poorly, and this region of the brain is important for that behavior, then shouldn't there be clear, interpretable differences in the neural data?

We thank the Reviewer for this comment. We have added a new Supplementary Figure 2, in which we present the data for both monkeys separately in the revised manuscript. Comparing the two datasets however, we see more commonalities related to the significant learning in both monkeys than differences that might be related to their different levels of learning. We have therefore decided to show the different datasets transparently in the new Supplementary Figure 2, but to stay on the conservative side in our interpretations.

- How should we look for these differences? A number of recent papers in mice have uncovered a large body of data showing that during the deliberation period, when the animal is interpreting a sensory stimulus (often using the whisker system), there is ramping activity in a principal component space among neurons that contribute to the decision. This ramping activity is present (in the PCA space) in the motor areas of the cortex, as well as in the medial and lateral cerebellar nuclei. Perhaps a similar computational approach would benefit the current manuscript.

We also appreciate this point. We have done the principal component analysis accordingly, and we indeed do find the ramping activity in several components of the dentate activity of both monkeys (Mi and Mo). We have now added a new Supplementary Figure 3 with the first three components of both correct and incorrect trials for Mi and Mo, highlighting their potential contribution.

- What is the hypothesis that is being tested? That is, what do you think might be the function of this region of the cerebellum in this task? It seems to me that we are not entirely in the dark, as previous literature on mice decision-making tasks has produced a reasonable framework: the deliberation period coincides with ramping activity in many regions of the frontal lobe and the cerebellum. Indeed, the ramp in the cerebellum appears to be a necessary condition for the ramp to be present in the frontal lobe. Thus, we should see such ramping activity in this task in the dentate. When the monkey makes the wrong choice, the ramp should predict it. If you don't see the ramping activity, then it is possible that the hypothesis is wrong, or that you are not recording from the right place.

It is indeed one of our specific hypotheses that the dentate cells can be direction-selective for the preparing cognitive component and/or sensorimotor response. We provide evidence that this hypothesis may be correct when we analyze the regular time response curves (see Figure 2 and the new Supplementary Figure 2 where the data of both monkeys are now presented separately). Moreover, we have now verified this by analysing the ramping curves of PCA space (new Supplementary Figure 3) and firing frequency of DN neurons that modulated upon presentation of the C-stimulus (new Supplementary Figure 4). These figures and findings are now referred to in the main text.

- As this is a difficult task that depends on the ability of the animals to understand the meaning of the cues, it is quite concerning that one of the monkeys performed poorly, particularly in the early sessions. Notably, the disparity between the two subjects is rather large: one monkey at the start of the recordings achieved a performance that was much better than the second monkey did at the end of the recording sessions. You highlighted the differences in performance in Figure 1D and mentioned that you started recording once the animals reached 60% performance. However, this did not make sense to me as the performance of Mi even after the final day of recording did not reach the performance of Mo on the first day of recording. Thus, in contrast to Mo, Mi appeared to be not ready for the task when the recording began.

We understand this point. However, please note that the learning performance of the monkeys concerned retraining sessions after they had had several weeks of vacation. So, even though it is correct that one of the two monkeys had a very good consolidation and started already at a relatively high level on the first retraining session, the other one also started and ended at a level above chance level (the y-axis starts at 0.5). We now highlight this point better in the Results section.

- One objective of having two monkeys is to illustrate that what is true in one animal is also true in the other. In some figures, you show that the neural data are significantly different, while in others you combine them into one. Thus, are you confident that the neural data across the animals should be combined, as you have done in Figure 2? Perhaps you can use the large differences in performance as a source of variance to find meaning in the neural data.

This is a valid question; as highlighted above, we have now addressed this point in the new Supplementary Figure 2, where the data for both monkeys are presented separately. Given the sample sizes and level of variances, it is in general difficult to draw conclusions about the potential differences and contributions, but the data are sufficiently transparent to observe common trends. With regard to linking differences in the neural data to the differences in performance level, please also consider Figure 4, the new Supplementary Figure 3 (with the ramping PCA component) and new Supplementary Figure 4 (with the additional analysis of the ramping activity of DN neurons that modulated upon presentation of the C-stimulus), which suggests that the ramping stage of Mo starts before that of Mi. This difference highlights the possibility that injecting accelerations of the simple spike modulations of Purkinje cells in the cerebellar hemispheres into the complex of cerebellar nuclei may be instrumental in improving the performance of responses to covert attention, akin to what has been shown for the impact of Purkinje cells of the vestibulocerebellum on eye movement responses to vestibular stimulation (De Zeeuw et al. 1995, J Neurophysiol). This possibility is now also raised in the Discussion.

- How do we know that these neurons, or even this region of the brain, contribute to this task? When a new task is introduced, the contributions of the region of the brain that is being studied are usually established via some form of manipulation. This question is particularly relevant here because the two subjects differed markedly in their performance, yet in Figure 3 you find that a similar percentage of neurons are responding to the various elements of the task.

We appreciate this question. As highlighted above, we are refraining from showing our muscimol manipulation (3 ml of 5 μg/ml muscimol, Sigma Aldrich), as it only concerns 1 successful dataset and 1 control experiment. We hope to replicate this reversible lesion experiment in the future and publish it when we have full new datasets of at least two monkeys available. As explained above, for this paper we have sacrificed both monkeys following a timed perfusion, so as to have similar survival times for the transport of the neuro-anatomical tracer involved.

- Behavior in both animals was better when the gap direction was up/down vs. left/right. Is this difference in behavior encoded during the time that the animal is making a decision? Are the dentate neurons better at differentiating the direction of the cue when the gap direction is up/right vs. left/right?

These data have now been included in the new Supplementary Figure 2; we did not observe any significant differences in this respect.

**Reviewer #2:**
- The authors trained monkeys to discriminate peripheral visual cues and associate them with planning future saccades of an indicated direction. At the same time, the authors recorded single-unit neural activity in the cerebellar dentate nucleus. They demonstrated that substantial fractions of DN cells exhibited sustained modulation of spike rates spanning task epochs and carrying information about stimulus, response, and trial outcome. Finally, tracer injections demonstrated this region of the DN projects to a large number of targets including several known to interconnect the visual attention network. The data compellingly demonstrate the authors' central claims, and the analyses are well-suited to support the conclusions. Importantly, the study demonstrates that DN cells convey many motor and nonmotor variables related to task execution, event sequencing, visual attention, and arguably decision-making/working memory.

We thank the Reviewer for this positive and constructive feedback.

- The study is solid and I do not have major concerns, but only points for possible improvement.

We thank the Reviewer for this positive feedback.

- A key feature of this data is the extended changes/ramps in DN output across epochs (Figure 2). Crudely, this presents a challenge for the view that DN output mainly drives motor effectors, as the saccade itself lasts only a tiny fraction of the overall task. Some discussion of this dichotomy in thinking about the function(s) of the cerebellum, vis a vis the multifarious DN targets the authors demonstrate here, etc., would be helpful.

We agree with the Reviewer and we have expanded our Discussion on this point, also now highlighting the outcome of the new PCA analysis recommended by Reviewer 1 (see the new Supplementary Figure 3).

- A high-level suggestion on the data: the presentation of the data focuses (sensibly) on the representation of the stimulus and response epochs (Figures 2-3). Yet, the authors then show that from decoding, it is, in fact, a trial outcome that is best represented in the population (Figure 4). While there is nothing 'wrong' with this, it reads slightly incongruously, and the reader does a bit of a "double take" back to the previous figures to see if they missed examples of the trial-outcome signals, but the previous presentations only show correct trials. Consider adding somewhere in the first 3 main figures some neural data showing comparisons with incorrect trials. This way, the reader develops prior expectations for the outcome decoding result and frame of reference for interpreting it. On a related note, the text contains an earlier introduction of this issue (p24 last sentence) and p25 paragraph 1 cites Figure 3D and 3E for signals "related to the absence of reward" - but the caption says this includes only correct trials?

We thank the Reviewer for bringing up these points. We have addressed the textual suggestions. Moreover, we have done the PCA analysis suggested by Reviewer 1 for both the correct and incorrect trials (see Supplementary material).

- P29: The discrepancy in retrograde labeling between monkeys (2 orders of magnitude): I realize the authors can't really do anything about this, but the difference is large enough to warrant concerns in the interpretation (how did the tracer spread over the drastically larger area? Isotropically? Could it cross more "hard boundaries" and incorporate qualitatively different inputs/outputs?). A small discussion of possible caveats in interpreting the outcomes would be helpful.

We fully agree with this comment. As highlighted in the text, in both monkeys we first identified the optimal points for injection in the dentate nucleus electrophysiologically and we used the same pump with the same settings to carry out the injections, but even so the differences are substantial. We suspect that the larger injection might have been caused by an air bubble trapped in the syringe or a deviation in the stock solution, but we can never be sure of that. We have added a potential explanation for the caveat that might have played a role.

- And a list of quick points:

We have addressed all points listed below; we want to thank the Reviewer for bringing them up.

P3 paragraph 2 needs comma "in daily life,".P4 paragraph 2 "C-gap" terminology not previously defined.P4 paragraph 2 "animals employed different behavioral strategies". Grammatically, you should probably say "each animal employed a different behavioral strategy," but also scientifically the paragraph doesn't connect this claim to anything about the DN (whereas, e.g., the abstract does make this connection clear).P5 paragraph 1 "theca" should be "the".P6 paragraph 1 problem with ignashenkova citation insert.P10 paragraph 1 I think the spike rate "difference between highest and lowest" is not exactly the same as "variance," you might want to change the terminology.P10 paragraph 1 should probably say "To determine if a cell preferentially modulated".P10 paragraph 1 last sentence the last clause could be clearer.P17 paragraph 2 should be something like "as well as those by Carpenter and..."?P20 caption: consider "...directionality in the task: only one C-stim...".P20 caption: consider "to the left and right in the [L/R] task...to the top/bottom in the [U/D] task".Fig1E and S1 - is there a physical meaning of the "weight" unit, and if none, can this be transformed into a more meaningful unit?P21 paragraph 1 consider "activity was recorded for 304 DN neurons...".P21 paragraph 1 "correlations with the temporal windows" it's not clear how activity can "correlate" with a time window, consider rephrasing (activity levels changed during these time epochs, depending on stimulus identity).P21 paragraph 1 should be "by comparing the number of spikes in a bin...".P22 paragraph 2 "when we aligned the neurons to the time of maximum change" needs clarification. The maximum change of what? And per neuron? Across the population?P22 paragraph 2 "than that of the facilitating" should be "than did the facilitating units".P24 paragraph 1 needs a comma and rewording "Within each direction, trials are sorted by the time of saccade onset".P24 paragraph 1 should probably say "Same as in G, but for suppressed cells".P24 paragraph 2 should say "more than one task event" not "events".P24 paragraph 2 needs a comma "To fully characterize the neural responses, we fitted".P25 paragraph 1 should probably say "we sampled from similar populations of DN".P34 paragraph 3 consider rephrasing the sentence that contains both "dissociation" and "dissociate".P37 last line: consider "coordination of cerebellum and cerebral cortex *in* higher order mental..."?P38 paragraph 1 citation needed for "kinematics of goal-directed hand actions of others"?P38 paragraph 1 commas probably not needed "map visual input, from high-level visual regions, onto..."References

- Herzfeld D.J., Kojima Y, Soetedjo R, Shadmehr R (2018) Encoding of error and learning to correct that error by the Purkinje cells of the cerebellum. Nat Neurosci 21:736–743.

- van Es, D.M., van der Zwaag W., and Knapen T. (2019) Topographic Maps of Visual Space in the Human Cerebellum. Current Biol Volume 29, Issue 10p1689-1694.e3May 20.

- De Zeeuw CI, Wylie DR, Stahl JS, Simpson JI. (1995) Phase relations of Purkinje cells in the rabbit flocculus during compensatory eye movements. J Neurophysiol. Nov;74(5):2051-64. doi: 10.1152/jn.1995.74.5.2051.